# Characteristics and Outcome of Children with Renal Cell Carcinoma: A Narrative Review

**DOI:** 10.3390/cancers12071776

**Published:** 2020-07-03

**Authors:** Justine N. van der Beek, James I. Geller, Ronald R. de Krijger, Norbert Graf, Kathy Pritchard-Jones, Jarno Drost, Arnauld C. Verschuur, Dermot Murphy, Satyajit Ray, Filippo Spreafico, Kristina Dzhuma, Annemieke S. Littooij, Barbara Selle, Godelieve A. M. Tytgat, Marry M. van den Heuvel-Eibrink

**Affiliations:** 1Princess Máxima Center for Pediatric Oncology, 3584 CS Utrecht, The Netherlands; R.R.deKrijger-2@prinsesmaximacentrum.nl (R.R.d.K.); J.Drost@prinsesmaximacentrum.nl (J.D.); A.S.Littooij-2@umcutrecht.nl (A.S.L.); G.A.M.Tytgat@prinsesmaximacentrum.nl (G.A.M.T.); m.m.vandenheuvel-eibrink@prinsesmaximacentrum.nl (M.M.v.d.H.-E.); 2Department of Radiology and Nuclear Medicine, University Medical Center Utrecht/Wilhelmina Children’s Hospital, Utrecht University, 3584 CX Utrecht, The Netherlands; 3Division of Oncology, Cincinnati Children’s Hospital Medical Center, University of Cincinnati, Cincinnati, OH 45229, USA; James.Geller@cchmc.org; 4Department of Pathology, University Medical Center Utrecht, 3584 CX Utrecht, The Netherlands; 5Department of Pediatric Oncology & Hematology, Saarland University Medical Center and Saarland University Faculty of Medicine, D-66421 Homburg, Germany; Norbert.Graf@uks.eu; 6UCL Great Ormond Street Institute of Child Health, University College London, London WC1N 1EH, UK; k.pritchard-jones@ucl.ac.uk (K.P.-J.); Kristina.Dzhuma@gosh.nhs.uk (K.D.); 7Oncode Institute, 3521 AL Utrecht, The Netherlands; 8Department of Pediatric Oncology, Hôpital d’Enfants de la Timone, APHM, 13005 Marseille, France; Arnauld.Verschuur@ap-hm.fr; 9Department of Paediatric Oncology, Royal Hospital for Children, Glasgow G51 4TF, Scotland; Dermot.Murphy@ggc.scot.nhs.uk (D.M.); buburay@hotmail.com (S.R.); 10Pediatric Oncology Unit, Department of Medical Oncology and Hematology, Fondazione IRCCS Istituto Nazionale dei Tumori, 20133 Milano, Italy; filippo.spreafico@istitutotumori.mi.it; 11Department of Pediatric Hematology and Oncology, St. Annastift Children’s Hospital, 67065 Ludwigshafen, Germany; barbaraselle@hotmail.com

**Keywords:** renal cell carcinoma, pediatric, treatment, survival, outcome

## Abstract

Pediatric renal cell carcinoma (RCC) is a rare type of kidney cancer, most commonly occurring in teenagers and young adolescents. Few relatively large series of pediatric RCC have been reported. Knowledge of clinical characteristics, outcome and treatment strategies are often based on the more frequently occurring adult types of RCC. However, published pediatric data suggest that clinical, molecular and histological characteristics of pediatric RCC differ from adult RCC. This paper summarizes reported series consisting of ≥10 RCC pediatric patients in order to create an up-to-date overview of the clinical and histopathological characteristics, treatment and outcome of pediatric RCC patients.

## 1. Introduction

Renal cell carcinoma (RCC) is a rare pediatric renal malignancy, accounting for 3.5% of all renal neoplasms in children aged 0–14 years [1,2,3,4,5,6,7,8,9]. However, RCC accounts for 70% of the renal cancers in children aged 15–19 years, with rapidly rising incidences after the age of 7 years to 50% of renal tumors by age 14 years [9]. Knowledge of clinical characteristics, outcome and treatment recommendations is often based on retrospective studies, which mainly describe small study populations, and the insights obtained from adult RCC. In contrast to the pediatric setting, RCC is the most common renal malignancy in adults. Recent studies have suggested that the epidemiological and histological characteristics of pediatric RCC differ from their adult counterparts [2,3,4,5,10,11]. Since 2004, this has been acknowledged by the WHO, which officially classified the translocation-type RCC (*MiT*-RCC), representing the predominant subtype of RCC in pediatric and young adult patients, as a specific entity [12,13,14]. *MiT*-RCC is characterized by translocations involving the *TFE3* gene located on chromosome Xp11.2 and less frequently the *TFEB* gene on 6p21, representing translocations of the *microphthalmia transcription factor* (*MiT*) *family* genes [15,16]. However, the prognostic value of these different histological subtypes and whether they can be used to guide therapy remains debatable [17,18,19]. 

Whenever radical nephrectomy (RN) is feasible for localized pediatric RCC, surgery is the most effective treatment [4]. Furthermore, nephron-sparing surgery (NSS), especially in low-volume tumors, can have clinical advantages in pediatric RCC to preserve renal function, while guaranteeing oncological safety [2,7,18]. In general, it is a challenge to differentiate pediatric RCCs from Wilms tumors at presentation based on imaging features [20]. The prognostic value of lymph node dissection (LND) and positivity of lymph nodes has been debated [3,4]. Reported 5-year survival rates vary between 60% and 85%, with superior survival rates for non-metastatic disease [3,21]. 

The aim of this review is to provide an up-to-date overview of the clinical characteristics, histology, treatment and outcome of all reported pediatric RCC series.

## 2. Methods

For this narrative review, the PubMed and Embase databases were searched in December 2017 (and updated in December 2019) using the main search terms ‘renal cell carcinoma’, ‘pediatric’ and ‘adolescent’ (Appendix A: Search strategy in PubMed and Embase/Medline). Cross-reference and citation check of included papers were executed using Scopus.

We included studies that (1) contained at least ≥10 well-described children or young adolescents ≤ 25 years at diagnosis with RCC, (2) were original articles, (3) were written in English language, and (4) were available in full text. After removal of duplicates, 7294 articles were included (Figure 1). After screening based on title and abstract, the remaining 72 articles were selected for full-text screening, of which 24 articles could be included based on the above criteria. After extensive exploration of the manuscripts, it became clear that more than one of these studies included overlapping patients. Selection of articles for exclusion and inclusion was pursued with support from leading colleagues from Europe (MMvdHE and GAMT) and the United States of America (JIG) based on study group, center and period of inclusion indicated in the articles (Appendix A: Transparency regarding patients in the articles identified after title and abstract screening using the in- and exclusion criteria).

Included studies used the Modified Robson Staging System and the TNM classification system [22,23,24,25,26]. Despite the fact that these two methods are comparable to a certain extent, stage groups could not be compared entirely (Appendix A: Stage grouping of the Modified Robson Staging System and the TNM Staging System of RCC). Furthermore, the TNM Staging System has been modified over the years. The 1997, 2002 and 2010 versions report, among other differences, varying interpretations of tumor size and multiple positive lymph nodes (Appendix A: Stage grouping of the Modified Robson Staging System and the TNM Staging Systems; Appendix A: TNM Staging Systems of RCC) [27,28,29,30]. Hence, in cases where overall staging was missing, tumor stage was assigned according to the used TNM Staging System and information present in the article [22,23]. Treatment approach was analyzed together despite the difference in staging systems.

Histological classification was described including *TFE* testing for *MiT-RCC*, following the WHO 2004 or 2016 classification, or without *TFE* testing, following previous classifications [12,31]. When *TFE* testing was performed, the method for testing (i.e., immunohistochemistry, karyotyping, and FISH) was, where possible, retrieved.

Original studies reported surgical treatment as radical nephrectomy (RN), simple nephrectomy (SN), partial nephrectomy (PN) and nephron-sparing surgery (NSS). For our review, we categorized RN and SN as a ‘total tumor nephrectomy’ and PN and NSS as a ‘partial tumor nephrectomy’. Since a review of imaging characteristics of pediatric RCC would require a different search and screening strategy, sporadic data on radiological features were not included in the tables and figures. Further, an analysis of germline mutations and/or syndromes associated with pediatric RCC would require a different search and screening strategy, resulting in no systematically reported data of these entities in this study.

## 3. Results

We identified 24 pediatric RCC series (Table 1). Among them, there were nine studies based on (mostly national) multicenter registries/databases (Appendix A: Studies based on multicenter registration and/or databases) [2,5,7,8,13,32,33,34,35]. The other series consisted mainly of single- and multicenter studies. After evaluation of the content of the 24 manuscripts, 8 papers were excluded because of high likelihood of overlap (nr. 17–24) [2,7,18,19,32,33,36,37]. As a result, sixteen studies without potential overlap were included for overall analysis (nr. 1–16) [3,5,6,8,13,14,17,34,35,38,39,40,41,42,43,44] (Figure 1, Table 1). In order to provide transparency of all manuscripts that met the inclusion criteria and the selection process, we kept the eight excluded studies visible, but separate (Table 1). When analyzing survival data according to tumor stage, we considered all 24 series, since these data were lacking in the majority of the 16 originally included papers.

### 3.1. Clinical Characteristics

The 16 included articles provided data on 551 pediatric patients with RCC (Figure 1, Table 1). The reported median and mean age ranged from 8 to 17 years and 9 to 17 years, respectively, with an overall age range of 1–28 years (Table 1). Altogether, 255 male and 296 female pediatric RCC patients were described in detail. Tumor localization was reported in 229 children (112 left sided and 114 right sided) (Table 1). Three patients with a bilateral pediatric RCC were reported [5,34]. Four young adults > 21 years were included in the study of Cajaiba et al. [13].

### 3.2. Tumor Stage

Staging was reported according to the mentioned TNM staging systems in 11 studies (Table 1). Stage distribution included 216 patients with stage I/II, 130 patients with stage III and 88 patients with stage IV tumors (Table 1). Three studies using the Modified Robson Staging System reported 31 stage I/II tumors, 15 stage III tumors and 27 stage IV tumors (Table 1). One study did not specify the stage of their pediatric RCC cases [35]. In 1 study, including 17 cases, staging was performed following an alternative co-operative group staging method not comparable with the other two staging systems [44]. Altogether, tumor stage following the TNM and Modified Robson Staging System was available for 510 patients, resulting in 247/510 (48%) cases of stage I/II, 145/510 (28%) of stage III and 115/510 (23%) of stage IV disease. In 52 patients, the sites of distant metastases were reported. The most frequent sites were the lungs (*n* = 26) and the liver (*n* = 15).

### 3.3. Presenting Symptoms and Diagnostic Features

In 13 studies, presenting symptoms were reported in detail for 250 patients (Table 2). The most common presenting symptoms were (gross) hematuria (*n* = 86), abdominal and/or flank pain (*n* = 79) and abdominal/palpable mass (*n* = 80). Only three patients presented with the classic triad of hematuria, abdominal/palpable mass and abdominal/flank pain. Sixty-eight patients were reported to present with more general symptoms, such as fever, weight loss, constipation and night sweats. In 31/250 patients, RCC was diagnosed as an incidental finding (Table 2).

Studies describing radiological characteristics of pediatric RCCs have focused mainly on *MiT*-RCCs. Chung et al. report hyperattenuation together with necrosis and calcifications on non-enhanced CT imaging [20]. Furthermore, *MiT*-RCC seems to show hyperintensity on T1-weighted, and hypointensity on T2-weighted imaging [45,46,47]. 

### 3.4. Histological Characteristics

So far, seven pediatric studies used the WHO 2004 classification system including *Xp11*/*MiT* translocation analysis, using predominantly immunohistochemical testing, and also fluorescence in situ hybridization (FISH) and/or next-generation sequencing (NGS) (Table 3) [5,6,13,14,38,40,42]. In these studies, 140/317 (44%) patients were classified as *MiT*-RCC, 80/363 (22%) as papillary type and 36/363 (10%) as clear cell-type RCC (Table 3). The other nine articles used earlier classification systems [3,8,17,34,35,39,41,43,44]. In studies lacking proper testing for *MiT*-RCC, 76/184 (41%) clear cell and 30/184 (16%) papillary type RCCs were reported, with 55/184 (30%) of cases remaining unclassified/not otherwise specified (Table 3) [8,41,44]. 

### 3.5. Treatment

In stage I/II pediatric RCC, surgery was the most frequently applied treatment modality. Detailed information on tumor resection was available in 14 studies, describing 282/296 patients undergoing surgery (Table 4). For these 282, a total tumor nephrectomy (RN and SN) was performed in 258 (91%) patients, and a partial tumor nephrectomy (PN and NSS) in 24 (9%) patients. Information on LND was available in 8 studies, reporting 105/175 patients who underwent LND (Table 4). There was a discrepancy in the number of studies reporting on LND (eight studies) and histological nodal status (nine studies), with six studies reporting both [5,14,34,39,40,42]. The majority of the included studies did not report detailed information on the number and anatomical location of lymph nodes. Only Estrada et al. described that 1 (out of 11) patient underwent second-look surgery for LND, because of a positron emission tomography (PET)-scan positive RCC residue (fluorine-18-2-fluoro-2-deoxy-D-glucose-avid lymphadenopathy) [43].

Fourteen studies reported treatment details for 215 patients (Table 5) [3,5,14,17,34,35,40,41,42,43,44]. In 54/215 (25%) patients with varying disease stages, all originating from The International Society of Paediatric Oncology–Renal Tumor Study Group (SIOP-RTSG) centers, pre-operative chemotherapy had been administered, mainly consisting of vincristine and actinomycin-D, which is the regimen adopted for children with presumed Wilms tumor [5,34]. Selle et al. showed that pre-operative chemotherapy did not achieve a significant reduction in tumor volume in 11/14 patients [5]. Indolfi et al. reported 5/5 patients with, and 9/11 patients without, pre-operative chemotherapy with stage I–II disease to be well and disease free, whereas 5/6 stage III–IV patients relapsed after both pre-operative and post-operative chemotherapy [34].

Moment of administration of pre- and post-operative treatment was often not reported and treatment details were therefore difficult to interpret. Hence, information on response to treatment was limited and no preferred treatment based on the time frame could be identified. Data on stage III disease (available for 23 patients) were more limited and variable than treatment for stage IV disease (available for 54 patients) (Table 6). Immunotherapy, mainly consisting of interleukin-2 and/or interferon-α, was the most frequently and earliest used treatment for stage III disease (*n* = 8/23), most specifically described in the study of Indolfi et al. (*n* = 7) [34]. Chemoradiation was administered in 5/23 cases, and radiotherapy in 6/23 cases.

Therapy for stage IV disease consisted predominantly of combinations of radiotherapy (*n* = 14), chemotherapy (*n* = 13) and immunotherapy (*n* = 12) (Table 6). One case series described three stage IV patients who received combined therapy (radiotherapy, chemotherapy and immunotherapy) pre-operatively or even instead of operation [17]. More recently, Ramphal et al. and Geller et al. used multimodality treatment and treated stage IV patients, using cytokines as well as oxaliplatin, gemcitabine, doxorubicin, 5-fluorouracil, irinotecan, axitinib and celecoxib (Table 6) [14,42]. 

### 3.6. Outcome

Detailed data on outcome are available from 14/16 studies (Table 1). Outcome data were described as event-free survival (EFS) and overall survival (OS) with a mean follow-up time ranging from 33 months to 20 years, which made it difficult to compare studies and to draw conclusions on change in outcome over time. Four studies described a 5-year OS, ranging from 57 to 92%, whereas three studies described 5 year EFS, ranging from 53 to 92% [5,8,35,42].

For analysis of outcome data according to tumor stage, three studies gave an overview of stage-dependent survival (Appendix A: Available included studies with information about survival according to tumor stage) [5,8,34]. A systematic review by Geller et al. analyzed tumor stage as a prognostic factor and described a decreased survival based on 243 patients when tumor stage increased using univariate analysis [3]. In the three included studies, stage I/II disease was found to be a favorable prognostic factor, whereas patients with stage III/IV disease represented the worst outcome group, in particular those with stage IV disease. Indolfi et al. showed a 20-year survival of 88.9% for low-stage (stage I–II), compared to 22.6% for high-stage (stage III–IV) disease (*p* = 0.0001) [34].

### 3.7. Prognostic Factors

In several studies, prognostic factors of pediatric RCC were described (Table 1). None of these studies identified independent prognostic factors through multivariate analyses. Stage III/IV disease was the most frequently described prognostic factor for outcome after univariate analysis (Table 1, Appendix A: Available included studies with information about survival according to tumor stage). Indolfi et al. reported low survival rates for patients with lymph node involvement (*n* = 41) (20-year EFS rate 50%) [34]. Geller et al. focused on the issue of N + M0 pediatric RCC and outcome in two papers, concluding that children and young adolescents have a favorable outcome compared to similarly staged adults, and recommended a surgery only approach until highly effective adjuvant treatments are identified [3,14]. One study with 46 cases showed that pediatric *MiT*-RCCs were significantly associated with a poorer survival rate than *TFE*-negative RCCs after univariate analysis (*p* = 0.035) [6].

## 4. Discussion

The present narrative review was conducted to present an up-to-date and comprehensive overview of available patient characteristics, administered treatment and outcome of pediatric RCC, by systematically summarizing all data in previous studies while correcting for possible double inclusion of patients. Overall, this effort reveals that, in comparison to adults, RCC in children seems to occur rarely. This is in line with recent population-based studies, showing RCC to be the predominant renal tumor type in (young) adolescents, whereas it occurs rarely up to the age of 14 years [9,48]. Moreover, the lack of data is partly based on the fact that pediatric RCC cases have not always been part of (international) renal tumor registries or protocols for children and adolescents.

The median reported age at diagnosis for pediatric RCC varies between 9 and 12 years [4,9]. The majority of the included studies show a median age of ≥10 years, although pediatric RCC cases at the age of 1 year have been reported. In adult RCC, there is a 2:1 male predominance [49,50,51,52]. Our review of pediatric RCC studies does not show convincing evidence for sex predominance, whereas recent population-based studies showed a female excess in adolescents over age 15 years [4,9,53,54]. With regard to tumor location, we expected and found equal distribution of left- and right-sided tumors in cases where this was specified (*n* = 229) (Table 1).

We identified hematuria, abdominal pain and mass as the most commonly reported presenting symptoms in pediatric RCC (Table 2). In predominantly young adult patients (*n* = 61), Tsai et al. reported, consistent with this pediatric review, hematuria as the most common symptom, followed by abdominal mass and pain [55]. Whereas a palpable abdominal mass is present in most patients with Wilms tumor, the percentage of patients presenting with hematuria and abdominal pain varies [18,44,56].

RCC has been associated with several pre-existing conditions. Argani et al. stated that renal insufficiency and cytotoxic chemotherapy in the medical history may predispose to the development of *Mit*-RCC [16]. Furthermore, syndromes such as von Hippel–Lindau (*VHL* gene) and tuberous sclerosis complex (*TSC1* and *TSC2* genes) can be associated with RCC [5,6,40]. These are among a variety of genes and syndromes associated with RCC, including Birt–Hogg–Dubé *(FLCN* gene) and hereditary leiomyomatosis and renal cell cancer (*FH* gene) (Appendix A: Genes and syndromes associated with RCC) [57,58,59,60]. Further analysis of the influence of these associations in pediatric RCC was beyond the scope of this paper.

Hyperintensity on T1-weighted and hypointensity on T2-weighted imaging of *Mit*-RCC seem to be distinctive characteristics since most tumors show opposite findings [20,46,47]. Furthermore, multiple studies have reported calcifications in >50% of pediatric *MiT*-RCCs, which seems to be less frequent in Wilms tumors [20,46,47,61,62,63]. Future innovations, such as radiological improvements (diffusion-weighted MRI) and cell-free DNA innovative research strategies could be relevant to further enhance discrimination of pediatric RCCs from other renal tumors at presentation [64,65].

With regard to tumor stage, approximately half of the patients were diagnosed with stage I/II disease. Nevertheless, the use of two different staging systems and the development of a new TNM staging system over time complicated the comparison of data, and could even result in a discrepancy in tumor stage. This stresses the importance of a uniform and updated TNM Staging System in future studies [37,66,67].

It has been acknowledged for several years now that pediatric RCC can be classified into a variety of histologic subtypes. These included papillary type RCC, RCC not otherwise specified, clear cell-type RCC and chromophobe RCC. A worldwide classification was, however, not available. Since the 2004 WHO classification system, *MiT*-RCC was officially recognized as a separate entity [12]. In addition, the recently discovered *succinate dehydrogenase* (*SDH*)-deficient RCC, *fumarate hydratase* (*FDH*)-deficient RCC and *anaplastic lymphoma kinase* (*ALK*)-rearranged RCC have been recognized as separate entities and are increasingly reported [13,68]. Fortunately, we have the 2016 WHO classification, which now enables worldwide comparison of registered cases in the future [31].

Overall, in published patients with available data, the *MiT*-RCC subtype was identified in 44%, which is in line with the previously stated conclusion that *MiT*-RCC is the most common subtype in pediatric and young adolescent patients under the age of 25 years [11,13]. Nevertheless, such percentages need to be considered with care, given most prior reports are limited by selection bias, thus not enabling accurate assessment of the percent of pediatric RCCs that are of certain histologic subtypes [69]. The study of Cajaiba et al., including >200 consecutive prospectively accrued centrally reviewed cases in the target age range in the USA during the years of accrual to AREN03B2, missed a minority of cases in the country, and therefore likely represents an accurate histologic breakdown [13]. When comparing series with and without included *TFE* testing, it is obvious that the percentage of pediatric patients with predominantly clear cell-type RCC and unclassified RCC has decreased after introduction of proper translocation testing. The correlation of differences in outcome and histological subtype has been extensively debated the past years. This further emphasizes the importance of analyzing *TFE* status, given the yet suspected relation of an aggressive clinical course and high-stage tumors in case of *MiT*-RCC [4,70,71].

In SIOP protocols, most children ≤10 years with renal tumors are pre-treated, based on the suspicion of having a Wilms tumor based on age, epidemiology and predominant incidence, without performing biopsy in the majority of cases [72]. This may delay nephrectomy and application of effective RCC treatment, and explains the high rate of pre-operative chemotherapy in several series [5,34,65]. So far, there seems no convincing beneficial effect of pre-operative chemotherapy, and no adverse effect of postponement of surgery [5,34].

Obviously, surgery is the most effective treatment regimen for pediatric RCC, resulting in cure in most patients with localized disease [34]. In small tumors, this can be pursued using NSS, to spare kidney function for the future [73,74]. Ramphal et al. described a relapse-free outcome for all four patients who underwent PN/NSS, only executed in stage I tumors [42]. Yang et al. stated in a recent review that RN was associated with fewer complications, whereas PN showed less reoperations and lower hospital mortality in adult RCC [75]. Conclusions on the role of nephrectomy in metastatic disease could not be drawn from this study, but remains an important focus for future studies.

Evidence for improved outcome after primary lymphadenectomy in pediatric RCC remains limited [4,76,77]. Lymphadenectomy, in addition to radical tumor nephrectomy in adult RCC patients, does not seem to significantly enhance survival compared to RN only [78,79,80,81]. However, LND may be beneficial in patients with locally advanced disease, when technically feasible [81]. Further, the preferred number of lymph nodes to remove remains an unanswered issue. Several studies showed and specified the impact of the presence of loco-regional positive lymph nodes on patient outcome [2,3,7,14,32,34]. Geller et al. stated that survival of adult N + M0 RCC is compromised, but not N + M0 pediatric RCC [3,14]. In current COG and SIOP protocols, lymph node sampling is performed in all pediatric kidney cancers. Furthermore, there might be a prognostic and therapeutic role for sentinel lymph node biopsy. However, evidence in children as well as in adults is lacking, especially in high-risk RCC [82,83].

With regard to treatment strategies other than surgery for more advanced stages, a variety of treatment regimens have been described. For patients with completely resected stage III tumors, there is still no evidence that adjuvant therapy provides a beneficial effect. There is, however, an urgent need for development of novel therapies, such as biologicals, for patients with metastasized or recurrent disease [19,37,42]. Administration of IFNα and high-dose IL-2 has led to complete or partial responses in adult RCC [84,85]. In the past, anecdotal objective responses in individual metastatic pediatric RCC have been described after administering a combination of IFN-α and/or IL-2 [7,86,87]. Still, strong evidence on the impact of immunotherapy in children is lacking [4,88].

Sunitinib is recommended in the UMBRELLA 2016 protocol for metastatic pediatric RCC as a first-line drug, since it significantly improved progression-free survival in metastatic RCC in adults [89,90]. Nevertheless, very limited outcome data in pediatric RCC are available and there is a lack of evidence for treatment of different histological subtypes of pediatric RCC [91]. Recently, the effect of various therapies has been described in a retrospective analysis in children and young adults, showing more objective responses with the Receptor Tyrosine Kinase (RTK) inhibitor sunitinib for *MiT*-RCC in comparison with sorafenib or mTOR inhibitors (temsirolimus or everolimus) [86]. In a study of Ambalavanan et al., 11 patients with stage IV *MiT*-RCC were treated with predominantly antiangiogenic drugs and biologicals, demonstrating a progression-free survival of approximately 5 months for sunitinib [19]. Recently, increased progression-free survival in advanced adult RCC was reported after a multiple kinase inhibitor (cabozantinib) or combination of a RTK inhibitor (axitinib) with a checkpoint inhibitor (pembrolizumab, avelumab), paving the way for more routine use of immune checkpoint inhibitors for RCC [92,93,94]. As adult RCC shows a different subtype distribution, the relevance for pediatric RCC, so far, is unclear [95]. Geller et al. studied the maximum tolerated and recommended dose of axitinib in children and adolescents with recurrent refractory solid tumors, which is now being included in advanced disease trials, in combination with nivolumab, in study AREN1721 for patients with *MiT*-RCC of all ages (NCT03595124) [1].

Therapeutic innovation is, for a large part, dependent on the availability of pre-clinical research models. Although several RCC cell lines have been established, they generally do not reflect the cellular and genetic heterogeneity of native tumors. Moreover, they do not cover the full spectrum of various genetic driver alterations found in childhood RCC. Lastly, their low establishment efficiency does not allow for development of individualized therapies. Development of new culture models, such as through organoid technology, allows for efficient establishment of 3D cultures from patient-derived tumor tissue. They typically can be long-term expanded while retaining key histological and genetic characteristics of the tissue they were derived from [96,97]. Importantly, multiple recent reports have demonstrated that tumor organoids have predictive value for drug sensitivity of patient tumors [98,99,100,101,102]. Calandrini et al. recently succeeded in generating organoid models from several *MiT*-RCCs [103]. These organoid models provide a novel platform for therapy development and studying fundamental cancer biology.

## 5. Conclusions

RCC is a rare renal tumor in children. There is evidence from recent large pediatric series that *MiT*-RCC and papillary type RCC are the most common subtypes. Recommendations for the optimal treatment approach for high-stage pediatric RCC tumors are under development, following target identification through novel biological models and targeted therapy approaches. It is of great importance that future studies focus on the revealed gaps of knowledge for pediatric RCC. Overall, cross-Atlantic prospective registration such as SIOP-RTSG UMBRELLA 2016 and AREN1721 by international collaboration will result in more specific knowledge for designing enhanced and effective treatment guidelines for subtypes of pediatric RCC.

## Figures and Tables

**Figure 1 cancers-12-01776-f001:**
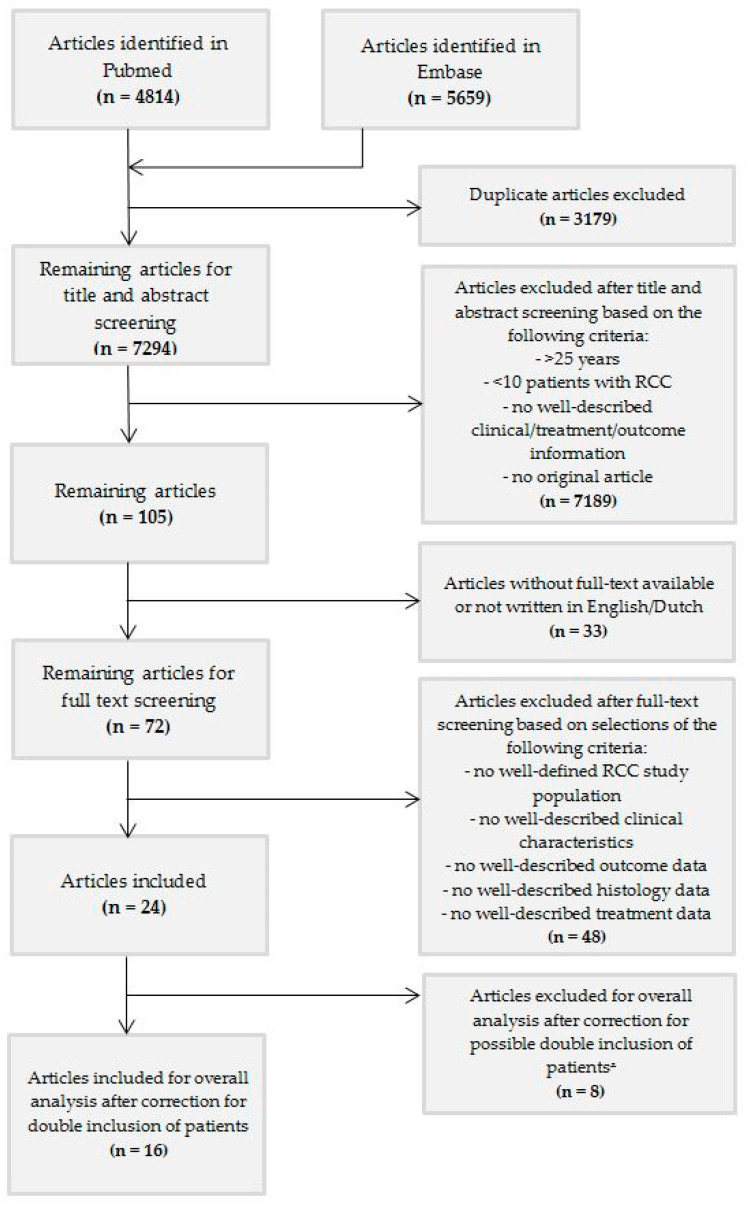
Flowchart. ᵜ These articles were used for additional results and information, if applicable.

**Table 1 cancers-12-01776-t001:** Overview of characteristics and outcome of pediatric renal cell carcinoma (RCC) based on all series describing ≥10 children and young adolescents.

#	Author (Year)	Country	Number of Patients	Gender		Tumor Side		Median(md)/Mean(mn) Age (Range) in Years		Staging System (Version)	Staging Group				Event-Free Survival (Mean/Median FU)	Overall Survival (Mean/Median FU)	Prognostic Factors
				Female	Male	Left	Right				I	II	III	IV			
Studies included for overall analysis																	
1	Cajaiba (2018) ᵜ [13]	USA	212	107	105	*NS*		NS (0.9–28)	*NS*	TNM ^±±^ (NS)	70	21	73	45	NS		NS
2	Kim (2015) [38]	South Korea	23	18	5	*NS*		10 (7–16)	md	TNM (NS)	12	4	5	2	5-year cancer-specific survival 85%		NS
3	Wang (2012) [39]	USA	12	6	6	7	5	11 (1–18)	mn	TNM^‡^ (NS)	5	1	5	1	11/12 (mean 33.0m)	12/12 (mean 33.0m)	NS
4	Rao (2011) [6]	China	46	26	20	22	24	16.5 (5–25)	md	TNM ^‡^ (2002)	9	15	18	4	25/39 (mean 55.5m)	26/39 (mean 55.5m)	Translocation type RCC
5	Silberstein (2009) [8]	USA	43	25	18	*NS*		15.4 (5–20)	mn	TNM (1997)	23		9	11	NS	5y & 10y 61%	Stage IV
6	Geller (2008) [14]	USA	11	8	3	5 ^±^	5 ^±^	16 (5–17)	md	TNM (1997)	3	1	1	6	NS		NS
7	Wu (2008) [40]	USA	13	4	9	*NS*		17 (9–23)	md/mn	TNM (2002)	6	2	4	1	8/13 (mean 5.6y)	9/13 (mean 5.6y)	NS
8	Varan (2007) [41]	Turkey	11	8	3	6	5	10 (3–16)	md	Modified Robson	3	1	3	4	60% (median 6.6y)	66.6% (median 6.6y)	NS
9	Selle (2006) [5]	Germany	49	25	24	20 ^±^	25 ^±^	10.6 (1.2–15.9)	md	TNM ^±±^ (1997)	17	11	4	8	5y 84.8%	5y 85%	Stage
						B: 2					V: 2						
10	Ramphal (2006) [42]	Canada	13	8	5	5	8	8 (2–18)	md	TNM (2002)	5	3	1	4	5y 92%	5y 92%	NS
11	Estrada (2005) [43]	USA	11	8	3	*NS*		14.7 (9.3–17.6)	mn	TNM ^‡^ (1997)	5	0	5	1	6/10 (mean 4.9y)	7/10 (mean 4.9y)	NS
12	Geller (2004) [3]	USA	13	6	7	4	9	12 (7–17)	md	TNM ^‡^ (1997)	2	1	5	5	9/13 (mean 8.6y)	10/13 (mean 8.6y)	Stage
13	Indolfi (2003) [34]	Italy	41	23	18	25	15	10.3 (1.5–17.9)	md	Modified Robson	18	1	12	9	20y 54%	20y 55%	Stage
						B: 1					V: 1						
14	Aronson (1996) [17]	USA	22	9	13	9	13	15.5 (3–21)	md	Modified Robson	7	1	0	14	NS	2y 45%, 5y 30%	NS
15	Chan (1983) [44]	Canada	17	10	7	*NS*		12 (1.3–20)	mn	CGS	3	9	5	0	8/17 (mean 9y)	8/17 (mean 9y)	NS
16	Dehner (1970) [35]	USA	14	5	9	9	5	9 (3–14)	mn	*NS*	*NS*	*NS*	*NS*	*NS*	5y 53%	5y 57%	Symptoms, histology, pseudocapsule, absence of vascular invasion
Total of included studies			551	296	255	112	114										
Studies excluded for overall analysis due to possibility of duplication of patients (Appendix A)																	
17	Ambalavanan (2019) [19]	USA	24	11	13	15	9	15 (3–27)	md	TNM (NS)	8	2	3	11	NS (all stage IV patients died from disease)		NS
18	Akhavan (2015) [32]	USA	515	257	258	*NS*		NS (0–21)	*NS*	*NS*	242	101	73	99	NS		Stage^ϒ^, government insurance, Asian race, no surgery
19	Geller (2015) [7]	USA	120	63	57	58 ^±^	60 ^±^	12.9 (1.9–22.1)	md	TNM ^±±^ (2002)	35	11	43	25	NS		NS
20	Rialon (2015) [2]	USA	304	145	159	*NS*		13 (9–16)	md	TNM ^±±^ (NS)	83	35	70	26	NS	1y 87%, 5y 70%	Size, stage, nodal status, no surgery
21	Indolfi (2012) [33]	Italy	14	7	7	6 ^±^	6 ^±^	13.0 (1.3–16.2)	md	Modified Robson	0	0	0	14	0% (median 7.5m)		NS
22	Baek (2010) [36]	South Korea	11	4	7	*NS*		12.7 (5–18)	mn	TNM (NS)	7	3	1	0	10/11 (mean 6.8y)	10/11 (mean 6.8y)	NS
23	Cook (2006) [18]	Canada	15	8	7	7	8	7.9 (2.5–18)	mn	Modified Robson	6	4	2	3	13/15	14/15	NS
24	Carcao (1998) [37]	Canada	16	11	5	10	6	9.6 (3–19)	mn	Modified Robson	3	0	7	6	NS	10/16 (mean 4.5y)	NS

ᵜ/ᵜᵜ Four patients were ≥21 year old; age, gender and staging group were not specified for these four patients, so they could not be excluded for the analysis. ^±^ Tumor side not specified for some included patients. ^±±^ Tumor stage not specified/missing for some included patients. ^‡^ TNM stage recoded to the 7th edition of American Joint Committee on Cancer (AJCC) Cancer Staging Manual. ^ϒ^ Akhavan et al. was the only study that proved stage to be an independent prognostic factor by multivariate analysis. B = bilateral; y = year; m = month; EFS = event-free survival; OS = overall survival; NS = not specified; TNM = TNM classification of malignant tumors, based on tumor, nodal status and metastases; FU = follow up; CGS = co-operative group staging.

**Table 2 cancers-12-01776-t002:** Presenting symptoms.

Author (Year)	Nr. ^ϒ^	Number of Patients	Triad ^1^ (%)	(Gross) Hematuria (%)	Abdominal/ Palpable Mass (%)	Abdominal/ Flank Pain (%)	Urogential Symptoms ^2^ (%)	Hypertension/ Renal Failure (%)	Associated with vHL Syndrome/Hirsutism (%)	General/Paraneoplastic Symptoms ^3^ (%)	Incidental Finding/ No Symptoms (%)
Kim (2015) [38]	2	23	-	8 (34.8%)	-	7 (30.4%)	-	-	-	-	8 (34.8%)
Wang (2012) [39]	3	12	-	2 (17%)	4 (33%)	3 (25%)	-	-	-	-	3 (25%)
Geller (2008) [14]	6	11	-	2 (18.2%)	3 (27.2%)	3 (27.2%)	1 (9.1%)	1 (9.1%)	-	2 (18.2%)	2 (18.2%)
Wu (2008) [40]	7	13	-	4 (30.8%)	1 (7.7%)	2 (15.4%)	-	1 (7.7%)	2 (15.4%)	-	2 (15.4%)
Varan (2007) [41]	8	11	-	4 (36.4%)	2 (18.2%)	3 (27.2%)	-	-	1 (9.1%)	1 (9.1%)	-
Selle (2006) [5]	9	49	NS	12 (30%)	22 (55%)	5 (12.5%)	3 (7.5%)	-	-	17 (42.5%)	6 (15%)
Ramphal (2006) [42]	10	13	1 (7.7%)	6 (46%)	3 (23%)	-	1 (7.7%)	-	-	3 (23%)	-
Estrada (2005) [43]	11	11	-	4 (36%)	1 (9%)	3 (27%)	-	-	-	-	4 (36%)
Geller (2004) [3]	12	13	-	10 (77%)	-	8 (61.5%)	-	-	-	11 (84.6%)	2 (7.7%)
Indolfi (2003) [34]	13	41	-	12 (29.2%)	9 (21.9%)	17 (42.5%)	-	-	-	8 (19.5%)	2 (4.9%)
Aronson (1996) [17]	14	22	2 (9.1%)	7 (31.8%)	17 (77.3%)	13 (59.1%)	2 (9.1%)	2 (9.1%)	-	11 (50%)	2 (9.1%)
Chan (1983) [44]	15	17	-	6 (35.3%)	10 (58.8%)	8 (47.1%)	-	4 (29.4%)	-	-	-
Dehner (1970) [35]	16	14	NS	9 (4.3%)	7 (50%)	8 (57.1%)	-	-	-	15 (107.14%)	-
Total ᵜ		250	3	86	79	80	7	8	3	68	31

^ϒ^ Article number referring to Table 1. ^1^ The classic triad consists of (gross) hematuria, abdominal/palpable mass and abdominal/flank pain. ^2^ (Chronic) pyelonephritis, dysuria, urinary retention. ^3^ Fever, weight loss, constipation, vomiting, nausea, anemia/pallor, malaise, polycythemia, hepatic dysfunction, night sweats and lumbar pain. ᵜ Number of symptoms was registered, meaning patients could have presented with more than one symptom. NS = not specified or not mentioned (since a lot of studies only report the ‘main’ or ‘most common’ clinical symptoms); vHL = von Hippel–Lindau; - = 0/not reported.

**Table 3 cancers-12-01776-t003:** Histological subtypes.

Author (Year)	Nr. *^ϒ^*	Number of Patients		Translocation Type (Number Positive/Tested) (Test)	Clear Cell	Papillary	Mixed Cell	Renal Medullary Carcinoma	Chromophobe	Oncocytoma	Granular Cell	Sarcomatoid	Renal Cell	FDH/ SDH	Post-NB	Other RCCs	Unclassified/Not Otherwise Specified
Cajaiba (2018) [13]	1	208	88/208	IHC/FISH/NGS	7	32	-	26	13	-	-	-	-	3/1	-	22 ^a^	16
Kim (2015) [38]	2	23	1/NS	NS	12	7	-	-	-	-	-	-	-	-	-	-	3
Rao (2011) [6]	4	46	19/46	IHC	9	17	-	-	-	-	-	-	-	-	-	-	1
Geller (2008) [14]	6	11	8/11	IHC	-	1	-	-	-	-	-	1	-	-	-	-	1
Wu (2008) [40]	7	13	6/13	IHC/FISH	5	1	-	-	-	-	-	-	-	-	-	-	1
Selle (2006) [5]	9	49	11/26	IHC	3	16	2	-	2	-	-	1	-	-	2	-	12
Ramphal (2006) [42]	10	13	7/13	IHC	-	6	-	-	-	-	-	-	-	-	-	-	-
Total		363	140/317		36	80	2	26	13	0	0	2	0	4	2	22	34
Wang (2012) [39]	3	12	-		4	6	-	-	2	-	-	-	-	-	-	-	-
Silberstein (2009) [8]	5	43	-		5	3	-	-	-	-	-	-	-	-	-	-	35
Varan (2007) [41]	8	11	-		3	4	-	-	1	-	-	-	-	-	-	-	3
Estrada (2005) [43]	11	11	-		6	5	-	-	-	-	-	-	-	-	-	-	-
Geller (2004) [3]	12	13	-		8	3	-	-	-	1	-	-	-	-	-	1 ^b^	-
Indolfi (2003) [34]	13	41	-		24	7	-	-	1	-	4	-	5	-	-	-	-
Aronson (1996) [17]	14	22	-		15	2	4	-	-	-	-	-	-	-	-	1 ^c^	-
Chan (1983) [44]	15	17	-		-	-	-	-	-	-	-	-	-	-	-	-	17
Dehner (1970) [35]	16	14	-		11	-	-	-	-	-	3	-	-	-	-	-	-
Total		184	-		76	30	4	0	4	1	7	0	5	0	0	2	55

*^ϒ^* Article number referring to Table 1; ^a^ tuberous sclerosis associated (9), ALK-rearranged (8), thyroid-like RCC (3), myoepithelial carcinoma (2), ^b^ neuroendocrine, and ^c^ adenocarcinoma. *NS* = not specified; IHC = immune-histochemical tested; FISH = interphase fluorescence in situ hybridization; NGS = next-generation sequencing; FDH = fumarate hydratase deficient; SDH = succinate dehydrogenase deficient; NB = neuroblastoma; - = 0/not reported.

**Table 4 cancers-12-01776-t004:** Surgical treatment and lymph node status.

Author (Year)	Nr. *^ϒ^*	Number of Patients	Type of Surgery		Lymph Node Dissection	Positive Nodal Status
			Total ᵜ	Partial ᵜᵜ		
Kim (2015) [38]	2	23	RN–18 (73.3%)	PN–5 (21.7%)	4 (17.4%)	*NS*
Wang (2012) [39]	3	12	RN–10 (83%)	PN–2 (17%)	8 (70%)	3 (25)
Rao (2011) [6]	4	46	RN–43 (93.5%)	PN–3 (6.5%)	NS	NS
Geller (2008) [14]	6	11	RN–11 (100%)	-	11 (100%)	3 (27.3%)
Wu (2008) [40]	7	13	RN–8 (61.5%)	PN–5 (38.5%)	1 (7.7%)	3 (23.1%)
Varan (2007) [41]	8	11	RN–10 (90.9%)		NS	NS
Selle (2006) [5]	9	49	RN–41 (83.7%)	PN–5 (10.2%)	46 (97.9%)	8 (16.3%)
Ramphal (2006) [42]	10	13	RN–8 (61.5%) SN–1 (7.7%)	PN–4 (30.8%))	8 (61.5%)	5 (38.5%)
Estrada (2005) [43]	11	11	RN–7 (63.6%) SN–4 (36.3%)	-	1 (9.1) ^‡^	5 (45.5%)
Geller (2004) [3]	12	13	RN–12 (92.3%)	-	12 (92.3%)	*NS*
Indolfi (2003) [34]	13	41	RN–15 (36.6%) SN–20 (48.8%)	-	15 (36.6%)	10 (24.4%)
Aronson (1996) [17]	14	22	RN–19 (86.4%)	-	NS	5 (22.7%)
Chan (1983) [44]	15	17	RN–17 (100%)	-	NS	9 (52.9%)
Dehner (1970) [35]	16	14	RN–14 (100%)	-	NS	NS
Total		296	Total: 258 RN: 233 SN: 25	Partial: 24	105	51

*^ϒ^* Article number referring to Table 1. ᵜ Total = total tumor nephrectomy, including RN and SN. ᵜᵜ Partial = partial tumor nephrectomy, including PN (and NSS). ^‡^ Only Estrada et al. reported one patient where second-look surgery for LND was done. *NS =* not specified; RN = radical nephrectomy; PN = partial nephrectomy; SN = simple nephrectomy; NSS = nephron-sparing surgery.

**Table 5 cancers-12-01776-t005:** Pre- and post-operative therapy for included patients.

Author (Year)	Nr. ^ϒ^	Number of Patients	(Neo-)Adjuvant Therapy (*n*) ^1^	Chemotherapy (CT)-*n* (%)		Radiotherapy (RT)-*n* (%)	Chemoradiation (CR)– *n* (%)	Immunotherapy (IT)– *n* (%)	Combination Therapy (COT)-*n* (%)	
				Pre-Op/IO	Post-Op	Post-Op	Post-Op	Post-Op	Pre-Op/IO	Post-Op
Geller (2008) [14]	6	11	CR(1), IT(1), COT(1)	-	-	-	RT + VBL–1 (9.1)	IL-2–1 (9.1)	-	IF-α + IL-2 + 5-FU and OX + CPT11 + GEM + DOX–1 (9.1)
Wu (2008) [40]	7	13	BMT+EXP(1), EXP(1), COT(1)	-	-	-	-	0	-	EXP (+BMT)–1 (7.7) IL-2 + CP + RT + SUN–1 (7.7)
Varan (2007) [41]	8	11	CT(1), CR(4), IT(1), COT(2)	-	ACT + VLB–1 (9.1)	-	RT + VC–4 (36.4)	IF- α–1 (9.1)	-	ACT + IF-α–1 (9.1) 5-FU + IF-α–1 (9.1)
Selle (2006) [5]	9	49	CT(21), RT(2), CR(2), IT(2), COT(2)	DOX/VCR/ACT–11 (22.4) VC–3 (6.1)	VC–7 (14.3)	RT–2 (4.1)	RT + CARBO + E + I + DOX–1 (2.0) RT + VCR + ACT–1 (2.0)	IL-2 + IF-α + 13-CA–2 (4.1)	-	IL-2 + IF-α + 13-CA + CAP–2 (4.1)
Ramphal (2006) [42]	10	13	RT(1), COT(1)	-	-	RT–1 (7.7)	-	0	-	CEL + VBL + IL-2–1 (7.7)
Estrada (2005) [43]	11	11	CR(3), IT(1)	-	-	-	RT + VCR + ACT–3 (27.3)	IL-2–1 (9.1)	-	-
Geller (2004) [3]	12	13	CR(2), COT(1)	-	-	-	CR–2 (15.4)	0	-	IL-2 + IF-α + 5-FU–1 (7.7)
Indolfi (2003) [34]	13	41	CT(64), IT(13), RT(7)	VCR + ACT(+VC)–40 (97.6)	VCR+DAC–13 (31.7) VC – 11 (26.8)	RT–7 (17.1)	-	IF-α–7 (17.1) IF-α + IL-2–4 (9.8) IL-2-2 (4.9)	-	-
Aronson (1996) [17]	14	22	CT(11), RT(10) CR(5), T(9), COT(8)	-	VCR + ACT–6 (27.3) DOX + MTX + CY–3 (13.6) B + VBL–2 (9.1)	RT–10 (45.5)	RT + VC–5 (22.7)	IF-α–5 (22.7) IL-2–3 (13.6) LAK–1 (4.5)	RT+/CT+/IT – 3 (13.6)	CT + IT–2 (9.1) IT + RT–2 (9.1) RT + CT + IT–1 (4.5)
Chan (1983) [44]	15	17	CT(1), RT(8), CR(3)	-	VBL–1 (5.9)	RT–8 (47.1)	RT + ACT + VCR + CY–3 (17.6)	0	-	-
Dehner (1970) [35]	16	14	RT(4)		-	RT – 4 (28.6)	-	0	-	-
Total		215	(195)	54	44	32	20	27	3	14

^ϒ^ Article number referring to Table 1. ^1^ Type of adjuvant therapy and number of patients receiving adjuvant therapy; CT = chemotherapy; RT = radiation-/radiotherapy; CR = chemoradiation; IT = immunotherapy; COT = combined therapy (predominantly IT + CT); B = biologicals; BMT = bone marrow transplantation; EXP = experimental; - = 0; Pre-Op = pre-operative; Post-Op = post-operative; IO = instead of operation; NS = not specified; MI = missing information; VC = various combinations; IO = instead of operation; IF-α = interferon-alfa; 5-FU = 5-fluorouracil; VBL = vinblastine; IL-2 = interleukin-2; OX = oxaliplatin; CPT11 = irinotecan; GEM = gemcitabine; DOX = doxorubicine; SUN = sunitinib; ACT = actinomycin-D; CP = cisplatin; CARBO = carboplatin; E = etoposide; I = ifosfamide; VCR = vincristine; 13-CA = 13-cis-retinoic acid; CAP = capecitabine; CEL = celecoxib; DAC = dactomycin; ADR = adriamycin; TAM = tamoxifen; MTX = methotrexate; CY = cyclophosphamide; B = bleomycin; LAK = lymphokine-activated killer cells; MPA = medroxyprogesterone acetate.

**Table 6 cancers-12-01776-t006:** Pre- and post-operative therapy registered for included patients with stage III and stage IV disease.

Author (Year)	Nr. *^ϒ^*	Staging System	Number of Patients-Stage III	Number of Patients-Stage IV	Adjuvant Therapy Totals	Treatment Patients Stage III (*n*)	Treatment Patients Stage IV (*n*)
Geller (2008) [14]	6	TNM	1	6	CR(1), IT(1), COT(1)	(0)	CR (RT+VBL) (1) IT (IL-2) (1) COT (IF-α + IL-2 + 5-FU and OX + CPT11 + GEM + DOX) (1) TU (1)
Wu (2008) [40]	7	TNM	4	1	BMT+EXP(1), EXP(1), COT(1)	(0)	EXP+BMT (1)
Varan (2007) [41]	8	Modified Robson	3	4	CT(1), CR(4), IT(1), COT(2)	CR (RT+CP) (1) CR (RT+ACT+MPA) (1) CT (ACT+VBL) (1)	CR (RT + ACT + VCR + MPA) (1) COT (ACT + IF-α) (1) COT (5-FU + IF-α) (1) IT (IF-α) (1)
Selle (2006) [5]	9	TNM	4	8	CT(21), RT(2), CR(2), IT(2), COT(2)	CT (ACT+VCR) (1) CR (RT+CARBO+E+I+DOX) (1)	COT (IL-2 + IF-α + 13-CA + CAP) (2) CT (DOX/VCR/ACT) (3) IT (IF-α) (1) CT (VC) (2)
Ramphal (2006) [42]	10	TNM	1	4	RT(1), COT(1)	(0)	RT (1) COT (CEL + VBL + IL-2) (1)
Estrada (2005) [43]	11	TNM ^‡^	5	1	CR(3), IT(1)	CR (RT+VCR+ACT) (1) IT (IL-2) (1)	CR (RT + VCR + ACT) (1)
Geller (2004) [3]	12	TNM ^‡^	5	5	CR(2), COT(1)	COT (IL-2+IF-α+5-FU) (1)	CR (2)
Indolfi (2003) [34]	13	Modified Robson	12	9	CT(64), IT(13), RT(7)	CT (6)	
						IT (7) RT (3)	IT (NS) RT (4)
Aronson (1996) [17]	14	Modified Robson	0	14	CT(11), RT(10) CR(5), IT(9), COT(8)	(0)	CT (VC) (8) IT (IF-α(+IL-2/LAK)) (8) RT (9) COT (RT+/CT+/IT) (3)
Chan (1983) [44]	15	CGS	5	0	CT(1), RT(8), CR(3)	CR (RT + ACT + VCR + CY) (1) RT (3) CT (VBL) (1)	(0)
Total			40	52	(191)	23	54

*^ϒ^* Article number referring to Table 1; ^‡^ TNM stage recoded to the 7th edition of American Joint Committee on Cancer (AJCC) Cancer Staging Manual; CGS = co-operative group staging; *NS* = not specified; CT = chemotherapy; RT = radiation-/radiotherapy; CR = chemoradiation; IT = immunotherapy; COT = combined therapy (predominantly IT + CT); B = biologicals; BMT = bone marrow transplantation; EXP = experimental. IF-α = interferon-alfa; IL-2 = interleukin-2; 5-FU = 5-fluorouracil; VBL = vinblastine; OX = oxaliplatin; CPT11 = irinotecan; GEM = gemcitabine; DOX = doxorubicine; ACT = actinomycin-D; MPA = medroxyprogesterone acetate; CP = cisplatin; CARBO = carboplatin; E = etoposide; I = ifosfamide; VCR = vincristine; 13-CA = 13-cis-retinoic acid; CAP = capecitabine; CEL = celecoxib; LAK = lymphokine-activated killer cells; CY = cyclophosphamide; MPA = medroxyprogesterone acetate.

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
