# Peer review of "Characteristics and Outcome of Children with Renal Cell Carcinoma: A Narrative Review"

_cancers, 2020, doi:10.3390/cancers12071776_

Round 1

Reviewer 1 Report

This is a comprehensive review of renal cell carcinoma at the pediatric age, a topic not frequently analyzed in the literature. Several specialties from international medical centers have been involved, what makes even more robust the information included on it. No major criticisms. The manuscript should be published as it is. No further comments.

Author Response

Point 1:

This is a comprehensive review of renal cell carcinoma at the pediatric age, a topic not frequently analyzed in the literature. Several specialties from international medical centers have been involved, what makes even more robust the information included on it. No major criticisms. The manuscript should be published as it is. No further comments.

Response 1:

We would like to thank the reviewer for his/her time, and the encouraging review of the manuscript.

Reviewer 2 Report

This review is very interesting and very well written. The authors must be complimented for summarizing all data in previous studies, correcting for possible double inclusion of patients. I particularly find the data of female patients being predominant in adolescence very interesting. Hormonal changes in adolescence may have a role, an unexplored field of research. Also, the discussion on the possible role of sentinel node biopsy in children is interesting, or on the possible role of immunotherapy. All fields for future research, that this paper highlights. The mention of organoid models to test drugs in the pre-clinicals eating is very appropriate, and could have been probably  expanded. Overall the review is very well done.

Author Response

Response to Reviewer 2 Comments

Point 1:

This review is very interesting and very well written. The authors must be complimented for summarizing all data in previous studies, correcting for possible double inclusion of patients. I particularly find the data of female patients being predominant in adolescence very interesting. Hormonal changes in adolescence may have a role, an unexplored field of research. Also, the discussion on the possible role of sentinel node biopsy in children is interesting, or on the possible role of immunotherapy. All fields for future research, that this paper highlights.

Response 1:

We would like to thank the reviewer for his/her time, and for the comments and encouraging words concerning the manuscript. We agree with the reviewer that the findings and apparent differences in adolescent and adult renal cell carcinoma (RCC) are very interesting. The hormonal changes in adolescence and their relation to the development of RCC may indeed be an interesting field of research in the future. The same applies for the subjects concerning the possible role of sentinel node biopsy in children with RCC, and the possible role of immunotherapy and other novel therapies.

Point 2:

The mention of organoid models to test drugs in the pre-clinical setting is very appropriate, and could have been probably expanded. Overall the review is very well done.

Response 2:

We would like to thank the reviewer for this suggestion, and have extensively considered an extension of this topic in the manuscript. We do believe we have included all important recent papers on organoid models and therapeutic innovation concerning pediatric RCC, including the study by Calandrini et al. (2020). Furthermore we hope the paragraph is a concise and informative overview of the recent developments in this area of research (line number 363-374). Given the extend of the manuscript and the topics discussed, an expansion of the discussion may not enhance the readability of the manuscript.

[103] Calandrini, C.; Schutgens, F.; Oka, R.; Margaritis, T.; Candelli, T.; Mathijsen, L.; Ammerlaan, C.; van Ineveld, R.L.; Derakhshan, S.; de Haan, S.; Dolman, E.; Lijnzaad, P.; Custers, L.; Begthel, H.; Kerstens, H.D.D.; Visser, L.L.; Rookmaaker, R.; Verhaar, M.; Tytgat, G.A.M.; Kemmeren, P.; de Krijger, R.R.; Al-Saadi, R.; Pritchard-Jones, K.; Kool, M.; Rios, A.C.; van den Heuvel-Eibrink, M.M.; Molenaar, J.J.; van Boxtel, R.; Holstege, F.C.P.; Clevers, H.; Drost, J. An organoid biobank for childood kidney cancers that captures disease and tissue heterogeneity. Nat Commun 2020, 11, 1310, doi:10.1038/s41467-020-15155-6.

Reviewer 3 Report

This is an interesting narrative review of children with renal cell carcinoma. Since it is a rare disease, the authors have used material from many centres treating more than 10 patients with this condition. The Pubmed and Embase databases were searched between December 2017 and December 2019. Of the more than 10,000 publications found in both databases, only 16 articles were qualified for analysis. Nevertheless, when analyzing survival data according to tumor stage, the Authors considered 24 series since these data were lacking in the majority of the 16 originally included papers. This seems to be a slight ambiguity regarding the process of data selection and analysis. The authors conclude that RCC is a rare renal tumor in children and that MiT-RCC and papillary type RCC are the most common subtypes. Recommendations for the optimal treatment approach for high-stage pediatric RCC tumors are under development.
Pending the results of prospective studies, the collected and discussed material concerning RCC in children seems to be one of the most reliable.

Author Response

Response to Reviewer 3 Comments

Point 1:

This is an interesting narrative review of children with renal cell carcinoma. Since it is a rare disease, the authors have used material from many centres treating more than 10 patients with this condition. The Pubmed and Embase databases were searched between December 2017 and December 2019. Of the more than 10,000 publications found in both databases, only 16 articles were qualified for analysis. Nevertheless, when analyzing survival data according to tumor stage, the Authors considered 24 series since these data were lacking in the majority of the 16 originally included papers. This seems to be a slight ambiguity regarding the process of data selection and analysis.

Response 1:

We would like to thank the reviewer for raising this important issue. We agree with the reviewer that considering all 24 identified series for the specific analysis of survival data according to tumor stage, while all the other data was only captured from the 16 ‘qualified’ articles, requires a critical re-evaluation. For this part of the manuscript and the additional supplemental Table S5: Available non-overlapping studies with information about survival according to tumor stage we analyzed the n=7 studies that included survival data according to tumor stage for potential double inclusion, shown in supplemental Table S2: Transparency regarding patients in the articles identified after title and abstract screening using the in- and exclusion criteria. This resulted in exclusion of the study of Akhavan et al. (2015) because of potential overlap of included patients with the study of Rialon et al. (2015). The results suggest that high-stage disease is an important prognostic factor for survival, by providing the most comprehensive overview of available, but non overlapping literature.

However, we do agree with the reviewer that we should follow the same process of data selection and analysis throughout the manuscript to prevent ambiguity and to ensure the quality of our method and results. We have therefore excluded the studies of Rialon et al. (2015) and Carcao et al. (1998) from Table S5: Available non-overlapping studies with information about survival according to tumor stage and have changed the title to Table S5: Available included studies with information about survival according to tumor stage (line number 225-226, line number 238, line number 391, Supplemental tables). In the revised manuscript, we have made corrections to the results section following these changes. We have changed the number of included studies in the table from 5 to 3, we have deleted the references to the studies of Rialon et al. (2015) and Carcao et al. (1998) and have deleted the data presented from the study of Rialon et al. (2015) in the results section (line number 224-233).

[32] Akhavan, A.; Richards, M.; Shnorhavorian, M.; Goldin, A.; Gow, K.; Merguerian, P.A. Renal cell carcinoma in children, adolescents and young adults: a National Cancer Database study. J Urol 2015, 193, 1336-1341

[2] Rialon, K.L.; Gulack, B.C.; Englum, B.R.; Routh, J.C.; Rice, H.E. Factors impacting survival in children with renal cell carcinoma. J Pediatr Surg 2015, 50, 1014-1018

[37] Carcao, M.D.; Taylor, G.P.; Greenberg, M.L.; Bernstein, M.L.; Champagne, M.; Hershon, L.; Baruchel, S. Renal-cell carcinoma in children: a different disorder from its adult counterpart? Med Pediatr Oncol 1998, 31, 153-158

Point 2:

The authors conclude that RCC is a rare renal tumor in children and that MiT-RCC and papillary type RCC are the most common subtypes. Recommendations for the optimal treatment approach for high-stage pediatric RCC tumors are under development.
Pending the results of prospective studies, the collected and discussed material concerning RCC in children seems to be one of the most reliable.

Response 2:

We would like to thank the reviewer for his/her time, and for the comments and encouraging words concerning the manuscript.

Reviewer 4 Report

This is an interesting review on paediatric renal cell carcinomas. The article is comprehensive, well organized and is of potential value in adding knowledge to the field. There remains to be many questions around paediatric renal cell carcinomas that are not answered and as explained by the authors' are beyond the scope of this review. MiT family translocation RCCs appear to be the most common subtype in the paediatric population. The authors' alluded to other possible RCCs that are associated with germline mutations/syndromic conditions as the SDH and FH deficient RCC. Other such syndromes include the Hereditary renal cell carcinoma, Birt-Hogg-Dube syndrome, tuberous sclerosis. Some of these RCCs are relatively newly described entities and can be challenging to diagnose without the appropriate molecular testing. It is possible that many of these entities are among those discussed in the cohort, however were not properly identified due to lack of the corresponding molecular testing. The article could benefit from a paragraph discussing these entities and their associated aberrant gene mutations.

Author Response

Response to Reviewer 4 Comments

Point 1:

This is an interesting review on paediatric renal cell carcinomas. The article is comprehensive, well organized and is of potential value in adding knowledge to the field. There remains to be many questions around paediatric renal cell carcinomas that are not answered and as explained by the authors' are beyond the scope of this review. MiT family translocation RCCs appear to be the most common subtype in the paediatric population.

Response 1:

We would like to thank the reviewer for his/her time, and for the comments and encouraging words concerning the manuscript. We agree with the reviewer that a lot of questions concerning pediatric RCCs remain unanswered. There appeared to be variety of topics to focus on concerning pediatric RCCs, in which we made the choice to report certain topics more detailed. However, we do hope this paper has also contributed to the identification of the gaps of knowledge, as well as important topics to focus on in future studies.

Point 2:

The authors' alluded to other possible RCCs that are associated with germline mutations/syndromic conditions as the SDH and FH deficient RCC. Other such syndromes include the Hereditary renal cell carcinoma, Birt-Hogg-Dube syndrome, tuberous sclerosis. Some of these RCCs are relatively newly described entities and can be challenging to diagnose without the appropriate molecular testing. It is possible that many of these entities are among those discussed in the cohort, however were not properly identified due to lack of the corresponding molecular testing. The article could benefit from a paragraph discussing these entities and their associated aberrant gene mutations.

Response 2:

We would like to thank the reviewer for this comment and agree that the role of associated germline mutations/syndromes in the development of RCCs was not discussed extensively in this manuscript. We also agree that this is a very interesting topic, in this context. Because a structured and complete overview of such data would require a different search strategy and different in- and exclusion criteria, we could not address this topic in this review. Since, as the reviewer already stated, a lot of these germline-/syndrome-associated RCCs are relatively newly described and knowledge of their role in pediatric RCC remains debatable, we felt this topic should be the focus of future research in order to provide a complete and reliable overview. We have therefore clarified this in the revised manuscript by adding the sentence ‘Also, an analysis of germline mutations and/or syndromes associated with pediatric RCC would require a different search and screening strategy, resulting in no systematically reported data of these entities in this study’ to the methods (line number 110-112).

The SDH- and FH-deficient RCC are molecular subtypes of RCC, included in the WHO 2016 classification. These molecular subtypes of RCC require specific testing in order to show their specific morphological appearance. Therefore, these genes play a role in the molecular subtype as well as they can play a role as germline mutations of RCC. As mentioned by the reviewer, these molecular subtypes were already included in the discussion (line number 292-294).

Nevertheless, we agree the article could benefit from an expansion of this subject, and more specific, associated genes and syndromes. We have therefore included an overview of all the genes associated with pediatric RCC in the Supplemental tables (Table S6: Genes and syndromes associated with RCC). Furthermore, we have expanded the paragraph discussing associated genes and syndromes, referring to Table S6, by writing ‘Furthermore, syndromes such as Von Hippel-Lindau (VHL gene) and tuberous sclerosis complex (TSC1 and TSC2 genes) can be associated with RCC.[5,6,40] These are among a variety of genes and syndromes associated with RCC, including Birt-Hogg-Dubé (FLCN gene) and Hereditary leiomyomatosis and renal cell cancer (FH gene) (Table S6: Genes and syndromes associated with RCC).[57-60] Further analysis of the influence of these associations in pediatric RCC was beyond the scope of this paper’ (line number 269-276, line number 391-392, Supplemental tables).

[5] Selle B.; Furtwangler, R.; Graf, N.; Kaatsch, P.; Bruder, E.; Leuschner, I. Population-based study of renal cell carcinoma in children in Germany, 1980-2005: more frequently localized tumors and underlying disorders compared with adult counterparts. Cancer 2006, 107, 2906-2914.
[6]
Rao, Q.; Chen, J.Y.; Wang, J.D.; Ma, H.H.; Zhou, H.B.; Lu, Z.F.; Zhou, X.J. Renal cell carcinoma in children and young adults: clinicopathological, immunohistochemical, and VHL gene analysis of 46 cases with follow-up. Int J Surg Pathol 2011, 19, 170-179.

[40] Wu, A.; Kunju, L.P.; Cheng, L.; Shah, R.B. Renal cell carcinoma in children and young adults: analysis of clinicopathological, immunohistochemical and molecular characteristics with an emphasis on the spectrum of Xp11.2 translocation-associated and unusual clear cell subtypes. Histopathology, 2008, 53, 533-544.

[57] Shuch, B.; Vourganti, S.; Ricketts, C.J.; Middleton, L.; Peterson, J.; Merino, M.J.; Metwalli, A.R.; Srinivasan, R.; Linehan, W.M. Defining early-onset kidney cancer: implications for germline and somatic mutation testing and clinical management. J Clin Oncol 2014, 32, 431-437.

[58] Haas, N.B.; Nathanson, K.L. Hereditary kidney cancer syndromes. Adv Chronic Kidney Dis 2014, 21, 81-90.

[59] Hol, J.A.; Jongmans, M.C.J.; Littooij, A.S.; de Krijger, R.R.; Kuiper, R.P.; van Harssel, J.J.T.; Mensenkamp, A.; Simons, M.; Tytgat, G.A.M.; van den Heuvel-Eibrink, M.M.; van Grotel, M. Renal cell carcinoma in young FH mutation carriers: case series and review of the literature. Fam Cancer 2020, 19, 55-63.

[60] Carlo, M.I.; Mukherjee, S.; Mandelker, D.; Vijai, J.; Kemel, Y.; Zhang, L.; Knezevic, A.; Patil, S.; Ceyhan-Birsoy, O.; Huang, K.C.; Redzematovic, A.; Coskey, D.T.; Stewart, C.; Pradhan, N.; Arnold, A.G.; Hakimi, A.A.; Chen, Y.B.; Coleman, J.A.; Hyman, D.M.; Ladanyi, M.; Cadoo, K.A.; Walsh, M.F.; Stadler, Z.K.; Lee, C.H.; Feldman, D.R.; Voss, M.H.; Robson, M.; Motzer, R.J.; Offit, K. Prevalence of germline mutations in cancer susceptibility genes in patients with advanced renal cell carcinoma. JAMA Oncology 2018, 19, 55-63.
